# Quantifying the Genetic Basis of Marfan Syndrome Clinical Variability

**DOI:** 10.3390/genes11050574

**Published:** 2020-05-20

**Authors:** Thomas Grange, Mélodie Aubart, Maud Langeois, Louise Benarroch, Pauline Arnaud, Olivier Milleron, Ludivine Eliahou, Marie-Sylvie Gross, Nadine Hanna, Catherine Boileau, Laurent Gouya, Guillaume Jondeau

**Affiliations:** 1INSERM U1148, 75018 Paris, France; thomas.grange@polytechnique.org (T.G.); melodie.aubart@aphp.fr (M.A.); Louise.Benarroch@inserm.fr (L.B.); Pauline.Arnaud@aphp.fr (P.A.); marie-sylvie.gross@inserm.fr (M.-S.G.); catherine.boileau@aphp.fr (C.B.); 2Service de Neurologie Pédiatrique, Hôpital Necker-Enfants malades, AP-HP, Université de Paris, Faculté de médecine Paris Centre, 75006 Paris, France; 3Centre national de référence pour le syndrome de Marfan et apparentés, AP-HP, Hôpital Bichat, 75018 Paris, France; langeois.m@chu-toulouse.fr; 4CHU Toulouse, 31300 Toulouse, France; 5Centre national de référence pour le syndrome de Marfan et apparentés, Département de génétique, AP-HP, Hôpital Bichat, 75018 Paris, France; nadine.hanna@aphp.fr; 6Université de Paris, UFR Médecine Paris Nord, 75010 Paris, France; 7Centre national de référence pour le syndrome de Marfan et apparentés, Service de cardiologie, AP-HP, Hôpital Bichat, 75018 Paris, France; olivier.milleron@aphp.fr (O.M.); ludivine.eliahou@aphp.fr (L.E.); 8INSERM U1159, 75018 Paris; laurent.gouya@inserm.fr

**Keywords:** marfan, heritability, modifiers, fibrillin

## Abstract

Marfan syndrome (MFS) is an autosomal dominant connective tissue disorder with considerable inter- and intra-familial clinical variability. The contribution of inherited modifiers to variability has not been quantified. We analyzed the distribution of 23 clinical features in 1306 well-phenotyped MFS patients carrying *FBN1* mutations. We found strong correlations between features within the same system (i.e., ophthalmology vs. skeletal vs. cardiovascular) suggesting common underlying determinants, while features belonging to different systems were largely uncorrelated. We adapted a classical quantitative genetics model to estimate the heritability of each clinical feature from phenotypic correlations between relatives. Most clinical features showed strong familial aggregation and high heritability. We found a significant contribution by the major locus on the phenotypic variance only for ectopia lentis using a new strategy. Finally, we found evidence for the “Carter effect” in the MFS cardiovascular phenotype, which supports a polygenic model for MFS cardiovascular variability and indicates additional risk for children of MFS mothers with an aortic event. Our results demonstrate that an important part of the phenotypic variability in MFS is under the control of inherited modifiers, widely shared between features within the same system, but not among different systems. Further research must be performed to identify genetic modifiers of MFS severity.

## 1. Introduction

Marfan syndrome (MFS (MIM#154700)) is a connective tissue disorder with an autosomal dominant inheritance and a prevalence of 1 in 5000 individuals worldwide. Clinical features of MFS involve the skeletal, cardiovascular, and ocular systems, as well as the skin, the lung, and the dura [1]. The life-threatening complication is aortic dissection, a surgical emergency, which occurs in dilated aorta [2]. However, the clinical presentation of MFS displays high variability in the symptoms of affected patients, as well as in their severity and age of onset. This variability is observed both among and within affected families and remains unpredictable [3]. In most patients, MFS is due to a mutation in the FBN1 gene [4], encoding fibrillin-1, the major component of microfibrils in the extracellular matrix [5]. Several studies have been performed to investigate genotype–phenotype correlations, but few correlations between MFS clinical presentation and FBN1 mutation types were found [6]: nonsense mutations have been associated with more severe phenotype [7,8], missense mutations involving a cysteine have been associated with increased risk of ectopia lentis [9], and mutations within exons 24–32 to neonatal MFS and severe presentations in adults [10]. However, high variability, even within families, demonstrates that MFS clinical presentation is mostly driven by modifiers unlinked to mutation type and/or location within the gene/protein.

Recently, we showed that the level of residual FBN1 expression in MFS patients with premature termination codons (PTC) was significantly associated with a few clinical features, mainly ectopia lentis [11]. Other genetic factors that may influence the phenotypic distribution are still to be determined, and it is impossible today to predict the clinical severity of the disease within a given patient [12]. This lack of accurate predictors of the clinical evolution of MFS is often a critical issue in genetic counseling. It leads to anxiety in patients with MFS and may be responsible for the inaccurate decision of not having children, because of the fear of severe skeletal features or an aggressive vascular disease. Environmental modifiers of phenotype may be recognized and controlled (hypertension or intensive sport activity for aortic dilatation). However, they cannot account for all the variability observed among patients, particularly in view of the early occurrence of some features, such as ectopia lentis. This strongly suggests that genetic modifiers of MFS determine clinical presentation. However, the fraction of the phenotypic variance under genetic control has never been quantified. Furthermore, MFS is a multisystem disorder, and genetic or environmental determinants might be different in the development of the ocular, cardiovascular, and skeletal phenotypes.

The purpose of this study was first to describe cross-correlations between MFS clinical features and identify groups of features that could have common genetic or environmental determinants. We then propose a model to estimate each clinical feature’s heritability and the influence of the major locus from phenotypic correlations between related individuals. Finally, based on the significantly higher prevalence of aortic events (dissection or surgery) in males, we looked at the influence of the sex of the transmitting parent on the severity of the cardiovascular features of the disease.

## 2. Materials and Methods

### 2.1. Study Population

This study was based on a cohort of 1306 MFS patients, collected since 1995 by the Centre de référence National sur les syndromes de Marfan et apparentés (National reference center on Marfan Syndrome and related diseases), in the Bichat university hospital (Paris, France). This ensures a high level of homogeneity and quality of data collection. The diagnosis was made according to the revised Ghent nosology [1], and only patients with an identified FBN1 mutation were included. As most MFS symptoms appear or quickly evolve during adolescence, patients below the age of 18 were excluded [13].

This research was performed in agreement with French bioethics law, and our institution (Greater Paris University hospital) was allowed to use all patient data for research. For this research, informed consent was obtained for all patients.

### 2.2. Clinical Features

Twenty-one typical MFS clinical features were considered: 3 are cardiovascular (thoracic aortic aneurysm (TAA), aortic dissection and aortic surgery), 2 are ophthalmological (ectopia lentis and ectopia lentis surgery), and 16 are found in various systems, notably the skeleton and are named “systemic features” (pectus deformity, scoliosis, facial dysmorphism, flat foot, protrusio acetabulae, ogival palate with dental overlap, flessum, hypermobility, spondylolisthesis, dolichostenomelia, “thumb sign”, “wrist sign”, arachnodactyly with both “wrist sign + thumb sign”, striae, hernia, and spontaneous pneumothorax). Aortic dilatation was expressed as a standard deviation of aortic root dilatation with respect to age, sex, and body surface area, calculated according to Campens et al. [14]. For patients who had had aortic surgery, it was calculated using the age and clinical data at the last follow up before surgery (if unknown, aortic diameter was considered to be 50 mm at the time of surgery, as 50 mm is the usual surgical threshold for MFS patients [15]). The frequency of each clinical feature and distribution of thoracic aortic dilatation across the study population is presented in the Appendix A.

### 2.3. Phenotypic Correlations

Phenotypic correlation coefficients were computed on S.A.G.E (Statistical Analysis for Genetic Epidemiology) release 6.3 [16], which provides weighted correlation estimates, adapting the Pearson correlation coefficient to take into account family trees topology [17], providing asymptotic standard errors [18].

To investigate the relationship between MFS clinical features, we computed cross-correlations between 19 MFS clinical features (for arachnodactyly, only the combination of both “wrist sign” and “thumb sign” was considered here, as the results were very similar to those for “wrist sign” alone or “thumb sign” alone), after correction for covariates (body-mass index, age, and sex).

### 2.4. Agglomerative Hierarchical Clustering

To confirm individual correlations between clinical features in terms of “clusters”, we applied agglomerative hierarchical clustering. Complete linkage clustering was chosen as linkage criteria, and the metric chosen was the Pearson correlation coefficient provided by S.A.G.E. and corrected for covariates. A 5% threshold was used to define the clusters.

### 2.5. Heritability

Due to the small sample of monozygotic and dizygotic twins in our cohort, we used a classical quantitative genetics model, adapted to the case of autosomal dominant inheritance. The phenotypic value P of a clinical feature was assumed to be given by:P=G+E
$$G=M_{m}+M_{spe}+A+D$$

*G* is the genotypic value and *E* the environmental effect. *A* and *D* represent the additive effect of genes and the dominance component, unlinked to the major locus FBN1, respectively. As MFS is considered to be mainly a monogenic disease with autosomal dominant inheritance, we added two other variables that account for the influence of the major locus FBN1. *M_m_* refers to the mean phenotypic difference between the MFS population and the general population. For a given phenotypic trait, its value will be the same for any individual diagnosed with MFS, whatever the mutation (and is 0 for non-Marfan individuals). However, different mutations might have different phenotypic consequences. This specific effect of a given mutation is taken into account by *M_spe_*, which has the same value for all affected members within a family (carrying the same mutation) but takes different values for different families with different FBN1 mutations.

Let *i* and *j* be two related individuals, *r* their familial relationship, and *d* its kinship degree. The phenotypic correlation between *i* and *j* for the considered clinical feature is then given by:(1)$$Corr\left(P_{i}\;,\;P_{j}\right)=\frac{Var\left(M_{spe}\right)+\frac{Var\left(A\right)}{2^{d}}+\frac{Var\left(D\right)}{4}1_{\left\{r=siblings\right\}}+Cov\left(E_{i},E_{j}\right)}{Var\left(P\right)}\;$$
where *Var*, *Cov,* and 1 are, respectively, the variance, the covariance, and the indicator function. Heritability (H2) is defined as the fraction of the phenotypic variance that can be explained by variations in genotypes:$$H^{2}=\frac{Var\left(G\right)}{Var\left(P\right)}$$

In the case of a monogenic disorder such as MFS, it can be further divided into heritability from the main locus (H2FBN1), and heritability due to genetic modifiers unlinked to FBN1(H2non-FBN1). The latter will be referred to as “non-FBN1 heritability”.
$$H^{2}=H^{2}{}_{FBN1}+H^{2}{}_{non-FBN1}\;H^{2}{}_{FBN1}=\frac{Var\left(M_{spe}\right)}{Var\left(P\right)}$$
$$H^{2}{}_{non-FBN1}=\frac{Var\left(A\right)+Var\left(D\right)}{Var\left(P\right)}$$

Only weak genotype–phenotype correlations are reported in the literature, as far as the FBN1 locus is concerned. Moreover, one of the few correlations that has been well established to date is the association between neonatal MFS in association with mutations within exons 24 to 32. However, these patients are likely never to reach the age of 18, and thus, are not recruited in our study. Therefore, global heritability can be assumed to be mainly driven by non-FBN1 heritability (H2 non-FBN1), and H2FBN1 will first be neglected. Double the phenotypic correlation between parents and offspring or between siblings has often been proposed as an estimator of heritability [19].
$$H^{2}\approx 2\;Corr\left(Siblings\right)$$
$$H^{2}\approx 2\;Corr\left(Parent-offspring\right)$$

However, siblings’ correlation is likely to overestimate heritability because of common environmental factors. On the other hand, parent–offspring correlation might slightly underestimate it as it does not account for the dominance component of heritability (as parents and offspring do not share allele couples, see Equation (1). Therefore, and to increase statistical power, we propose to use the mean of these two estimators.

### 2.6. Statistical Tests

Statistical tests with a *p*-value < 0.05 were considered to be significant. To assess correlations between clinical features, as multiple tests were performed simultaneously (171 tests), the significance threshold was adapted according to the Bonferroni correction (*p* < 3.0 × 10^−4^).

## 3. Results

### 3.1. Cross correlations between Clinical Features and Agglomerative Hierarchical Clustering

Out of the cohort of 1306 MFS patients with an identified FBN1 mutation, 999 patients over 18 years old were included in this study. They belonged to 674 independent families, with 1 to 13 affected individuals in each family. Patients were between 18 and 89 years old at the time of the last follow-up, with a median age of 38, and slightly more females than males (53%, non-significant difference). Frequency of clinical features are reported in Appendix A. We found strong correlations between clinical features within each affected system (skeletal, cardiovascular, and ocular) (Figure 1), some being expected (aortic dilatation and aortic surgery, for example). On the other hand, only one correlation between features belonging to different systems was found to be significant (arachnodactyly - aortic dilatation, Figure 1) Even with a significance threshold 10 times higher than the one given by Bonferroni (3.10-3), only three more intra-systemic correlations could be found (pneumothorax-facial dysmorphism, pneumothorax-aortic dilatation, and aortic surgery-acetabular protrusion), and might be explained by sampling fluctuation, with respect to the very high number of correlations computed (171 tests). Since either “wrist” sign or “thumb” sign provided similar results to the combination of both, only the latter was conserved, and referred to as “arachnodactyly”.

We also applied hierarchical clustering on clinical features to confirm these individual strong intra-systemic and weak inter-systemic phenotype correlations in terms of “clusters”. Complete linkage clustering was chosen as linkage criteria, and the metric chosen was Pearson’s correlation coefficient corrected for covariates (as above, Figure 2). A 5% threshold provided five different clusters of clinical features: one cluster grouped ocular features, another one cardiovascular features, while the three remaining clusters pooled various systemic features.

### 3.2. Estimate of Heritability of MFS Features

A total of 623 patients from 196 families with at least two affected individuals could be used to estimate heritability. We computed phenotypic correlations between first degree relatives for 21 MFS clinical features, after correction for age and sex (Table 1). Heritability estimates were deduced from these correlations for each clinical feature. For most clinical features, we found a strong familial correlation, suggesting high heritability. The highest one was found for ectopia lentis and ectopia lentis surgery (> 60%), which gave similar results. Heritability of most systemic features ranged from 40% to 60%. Only six features (spontaneous pneumothorax, hernia, flessum, aortic dissection, spondylolisthesis, and hypermobility) had low and non-significant heritability. Finally, heritability of thoracic aortic dilatation (TAA) was estimated to be 46% and heritability of other cardiovascular features appeared to be weaker (aortic surgery: 27%; aortic dissection: 21%) (Table 1, Figure 3).

### 3.3. H2 FBN1 and H2 non-FBN1 Heritability

We then estimated the fraction of heritability explained by variations at the main locus FBN1 (i.e., H2FBN1, that has been neglected) with two methods.

We first focused on families with a PTC in the FBN1 locus, as it provides a genetically homogeneous group, leading in most of the cases to a null allele (i.e., H2FBN1is null in this group so that Var(Mspe) = 0). We took advantage of this property to estimate H2FBN1 by subtracting H^2^ obtained in the subgroup of PTC FBN1 families only (estimating H2 non-FBN1) from H2 obtained in all FBN1 families (AF), for siblings (Sib, Equation (2)), and also for parent–offspring (PO, Equation (3)). Changing the study population modifies not only the numerator (the genetic variance) but also the denominator (the global phenotypic variance) of heritability. Forgetting this would lead to misconceptions in heritability studies [20]. Formally, we deduced from Equation (1) that (detailed proof of Equations (2) and (3) in Appendix A):(2)$$H^{2}{}_{FBN1}=\frac{Corr\left(Sib.\;AF\right)-Corr\left(Sib.\;PTC\right)}{1-Corr\left(Sib.\;PTC\right)}\;$$
(3)$$H^{2}{}_{FBN1}=\frac{Corr\left(PO\;AF\right)-Corr\left(PO\;PTC\right)}{1-Corr\left(PO\;PTC\right)}\;$$

We used the mean of these two values. Results are available in Appendix A. Unfortunately, low statistical power prevented us from obtaining a reasonable estimate for any specific clinical feature. A global look at all features shows, however, (1) that 16 estimates out of 22 are positive which is unlikely explained by chance (*p* < 0.05), (2) that the mean value of H2FBN1 among features was 4.4%, reflecting a low influence of variations at the main locus on the phenotype.

Another approach to investigate the influence of the major locus is to look at distant relatives. Distant relatives carry the same FBN1 mutation, but do not share more environmental or genetic factors than unrelated MFS patients. Therefore, if variations at the major locus contribute largely to the phenotypic variance, phenotypic correlations are expected to remain high for distant relatives. An unbiased estimate of H2FBN1 would be given by:$$H^{2}{}_{FBN1}=\underset{d\to \infty }{\lim}\frac{\frac{Var\left(A\right)}{2^{d}}+Var\left(M_{spe}\right)}{Var\left(P\right)}=\underset{d\to \infty }{\lim}Corr\left(P_{i},P_{j}\right)$$
where *d* is the kinship degree between relatives *i* and *j*. We calculated avuncular and cousins phenotypic correlations for one ocular (ectopia lentis surgery), one skeletal (arachnodactyly), and one cardiovascular (TAA) MFS feature (Table 2). At a high level of kindship (cousins, 3 degree of kindship), a very high correlation was found only for ectopia lentis surgery (R = 45%, 95% confidence interval [21%, 68%]; Table 2.

### 3.4. Influence of the Sex of the Transmitting Parent and “Carter Effect”

Finally, we looked at the influence of the sex of the transmitting parent on the cardiovascular phenotype and found the “Carter Effect” [21]. The effect refers to the existence of a polygenic threshold model with sex dimorphism that postulates the presence of a greater number of susceptibility polymorphisms in affected carriers of one sex as compared to the other. In keeping with the general agreement for the vascular disease [22], we found that 61% of patients with aortic surgery or dissection in the study population were males (*p* = 4.9 × 10^−5^, Figure 4A). Among MFS patients whose transmitting parent had had aortic surgery or dissection, we found that the prevalence of aortic surgery or dissection was significantly higher in the case of maternal transmission (29.3%) than in paternal transmission (9.8%, *p* = 0.018, Figure 4B). The number of males and females was similar in each group, and the result was confirmed in both male (non-significant) and female subgroups (*p* = 0.0085, Figure 4B). On the other side, patients of the most susceptible sex that have not developed the disease are likely to have and transmit fewer genetic susceptibility factors. Among MFS patients whose transmitting parent had not had aortic surgery or dissection, we found that the prevalence of aortic surgery or dissection was slightly lower in the case of a paternal transmission (14.9%) than a maternal transmission (18.3%, Figure 4C). This difference was not significant. However, there were nearly twice as many males as females in the group of paternal transmission (31 and 16, respectively), which is likely to have artificially increased the prevalence of severe aortic phenotype and, thus, decrease the difference between paternal and maternal transmissions. This was confirmed by a tendency toward less aortic surgery or dissection in the case of a paternal transmission in both male and female subgroups (Figure 4C). Finally, the results were consistent between subgroups: All other things being equal, in each category, the prevalence of aortic surgery or dissection was higher in males and was higher when the transmitting parent had had aortic surgery or dissection, as expected (Figure 4B,C).

This phenomenon, known as the “Carter effect”, provides a further argument to support the existence of inherited modifiers in the MFS cardiovascular phenotype.

## 4. Discussion

The clinical presentation of MFS displays great heterogeneity, both intra- and inter-familial. It makes the clinical diagnosis of MFS difficult and limits genetic counseling. Since 2010, diagnosis is made according to the revised Ghent nosology, which puts more weight on aortic root aneurysm and ectopia lentis [1,23].

The study of the phenotypic expression of MFS by two approaches (cross-correlations between common MFS clinical features and hierarchical clustering on clinical features) leads to similar conclusions. Both methods demonstrated a strong intra-systemic phenotype correlation contrasting with a very low inter-systemic correlation. If FBN1 was the main driver of MFS clinical presentation, all symptoms would be correlated. These results, therefore, suggest, first, that the FBN1 locus by itself is not the main driver of phenotypic presentation of individual symptoms despite the FBN1 gene mutation being a necessary condition for MFS to occur. They also suggest that features within the same system share common underlying determinants and, on the other hand, that the non-FBN1 genetic or environmental determinants of cardiovascular, ocular, and systemic features of MFS are likely to be different. It also highlights the importance of considering separately and attributing an independent value to the three main clinical blocks (heart, eye, and skeleton) in the clinical diagnosis of MFS (that is effectively done in the Ghent nosology) and that, finally, has an important impact on future studies aiming to identify genetic modifiers.

Altogether, heritability appears quite high, ranging from 40% to 60% for most of the signs despite the fact that most clinical features were quoted as a binary variable. Binary features are often assumed to reflect a latent continuous variable (called liability) such that patients with a liability above a given threshold will be diseased. The calculated heritability on the binary scale is then lower than heritability on the continuous liability scale, even more so for features whose prevalence is very high or very low [24]. Therefore, the heritability of some features might have been underestimated in our study. However, only six features (spontaneous pneumothorax, hernia, flessum, aortic dissection, spondylolisthesis, and hypermobility) had low and non-significant heritability. Interestingly, these six features are binary and also have the lowest prevalence in the study population, which reduces statistical power and decreases heritability estimates on the observed binary scale. It is also possible that some of these events rely on rare and non-heritable environmental factors.

Cardiovascular features (aortic dilatation, aortic surgery, aortic dissection), although well correlated in the phenotypic study, showed an important difference in heritability. While TAA was assessed by a continuous variable, both aortic surgery and dissection are binary features with relatively low prevalence (31.9% and 9.8%, respectively), which may have affected both correlation coefficient and statistical power [24]. Furthermore, these quite low heritability estimates for aortic surgery or dissection were also due to a very weak parent–offspring correlation in contrast to the high correlation between siblings. This discrepancy may reflect either dominance effects or a confounding effect of age. If there were a dominance effect, one would expect a similar discrepancy for TAA. Yet, siblings and parent–offspring correlations were comparable for TAA (23%). It is, therefore, more likely that these low parent–offspring correlations for aortic surgery and dissection reflect the effect of age, and we know that TAA is usually present at the age of 18 [13], whereas aortic surgery or dissection often occurs later. Moreover, prevention has improved greatly (life expectancy of MFS patients has increased by 30 years in the last decades [25]), leading to heterogeneity between “old” and “young” patients (with respect to the year of diagnosis and years of follow-up). Whatever the explanation, it can be assumed that the sibling correlation reflects heritability of these two clinical features with more accuracy than the parent–offspring correlation. This assumption would lead to 40% and 44% heritability for aortic surgery and dissection, respectively, similar to that of aortic dilatation (46%). It is interesting to note that despite the high heritability of these phenotypes, the identification of rare or common genetic modifiers remains a very difficult challenge. Our group, in a multiomic approach in a large cohort of 1070 clinically well-characterized FBN1marfan patients, identified nine modifier loci in the polygenic model [26]. None of them has allowed the identification of the responsible variant(s) with certainty, and the use of these results for genetic counseling is still far away. Few studies have measured the heritability of aortic aneurysm susceptibility. In particular, there are two Scandinavian studies using the measure of concordance between mono- and dizygotic twins from Swedish and Danish general population registries [27,28]. The phenotype studied was abdominal aortic aneurysm. The two studies give similar heritability results, 70% and 77%, respectively, and an influence of unshared environmental factors of 20% to 30%. These heritability values are significantly higher than our measurements for thoracic aortic aneurysm in Marfan patients, whereas environmental factors, such as smoking, are considered to be a determinant in the occurrence of abdominal aortic aneurysm. This result is all the more noteworthy since twin studies control for the influence of the shared environment, which intrafamilial correlation studies cannot do and, therefore, tend to overestimate heritability.

The genetic variability of clinical expression in this dominant disease may reflect both the influence of the FBN1 locus (H^2^ FBN1, reflecting genotype/phenotype correlations) and or the influence of yet unidentified loci independent of FBN1(H2 non-FBN1). This can be estimated through two approaches: 1) comparing the phenotypic correlations for families carrying a null allele mutation (indicating H2 non-FBN1 assuming that all PTC mutations have the same functional impact) with the phenotypic correlations for all MFS families (indicating H2 FBN1+ non-FBN1, i.e., H2). Due to low statistical power, we were not able to estimate robustly H2FBN1 for each phenotype. Nevertheless, 16 estimates out of 22 were positive, which probability under the null hypothesis (i.e., no specific influence of the FBN1 gene mutation) is only 4.7% suggesting a significant influence of the major locus on phenotypic variability (*p* < 0.05). The mean average value was H2FBN1 4.4%, indicating a significant but low influence of the major locus on the phenotype. This result was strengthened by the comparison of heritability between close and distant relatives. The influence of the FBN1 locus remains stable, whereas the influence of non FBN1 factors decreases proportionally to 2-d the amount of non FBN1 genetic factors shared. A very high correlation was found between cousins for ectopia lentis surgery (R = 45%, 95% confidence interval [21%, 68%]). Supposing that the variability at the FBN1 locus has no influence on the ocular phenotype (i.e., Var(*M_spe_*) = 0), a 21% correlation between cousins (lower bound of the confidence interval) would then lead to an absurd 168% H2 heritability (Corr(cousins) = Var(A)/8Var(P) = H2/8 as cousins share on average 1/8 of their genome) [19]. This demonstrates that variability at the FBN1 locus has a significant influence on the ocular phenotype of MFS patients, and is consistent with the literature. Faivre et al. [9] reported a higher probability of ectopia lentis for patients with a missense mutation substituting or producing a cysteine residue when compared with other missense mutations. Cysteine amino acids implicated in disulfide bonds are consensus amino acids in the calcium-binding EGF-like (EGF stands for Epidermal Growth Factor) domains of the fibrillin-1 protein [29] and are, therefore, critical to maintaining its conformation. Decreasing sample size for distant relatives led to fluctuating correlations not sufficiently precise to permit conclusions for TAA or arachnodactyly. Taken together, our results suggest an overall low influence of the major locus on phenotypic variability, except for the ocular phenotype. This combined effect of both the major locus and the rest of the genome could explain why global heritability (H2 FBN1 + H2 non-FBN1) is higher for the ocular system than for other features. There are few examples of studies measuring the heritability of different phenotypes associated with an autosomal dominant Mendelian disease. Sebbagh et al. were nevertheless able to measure the heritability of different phenotypes associated with neurofibromatosis type 1 in a study of phenotypic correlations in 275 multiplex French NF1 families. As in our study, patterns of familial correlations indicated a strong heritability with no apparent influence of the constitutional NF1 mutation [30].

Finally, it has been reported that males with MFS had a more severe cardiovascular phenotype than females [22] (in our cohort, 61% of patients with aortic surgery or dissection were males). Based on this observation, we looked for a “Carter Effect” [21] in the cardiovascular phenotype of MFS patients. This phenomenon is observed in diseases with polygenic susceptibility factors when there is a significant difference in the proportion of affected males and females. This sexual dimorphism suggests that in the protected sex (here females), stronger/more genetic determinants, in addition to the FBN1 gene mutation, are needed to develop the disease (here aortic surgery or dissection) than in the more susceptible sex. Half of these determinants are transmitted to the next generation, leading (for equally affected parents) to an imbalance in the prevalence of the disease (aortic surgery or dissection) depending on the sex of the transmitting parent: Females with an aortic event would transmit more susceptibility factors than males with an aortic event. Our results provide evidence for the “Carter Effect” in the cardiovascular phenotype of MFS patients. There is an increased risk of aortic surgery or dissection for children when the transmitting parent is a woman who had undergone aortic surgery or dissection. This is also an additional argument in favor of a polygenic model and for the strength of heritability (or possibly of multiple inherited environmental susceptibility factors) for MFS cardiovascular clinical variability. This observation should also be translated into clinical practice: Clinicians need to be warned that having a severely affected mother (or probably severely affected women in their family) confers an additional risk of severe cardiovascular features, even more so for male children, more susceptible.

The potential existence of inherited environmental factors shared among relatives and that would affect the phenotype is confounding in our estimation of heritability. However, it is probably limited as most clinical features have an early onset (which rather suggests a genetic influence).

## 5. Conclusions

In summary, our results demonstrate that an important part of the phenotypic variability in MFS is under the control of inherited modifiers. These modifiers appear to be widely shared between features within the same system (i.e., ophthalmology vs. skeletal vs. cardiovascular), but not among different systems. Therefore, studies limited to one system should be considered carefully. In particular, no extrapolation can be made from one system to another as far as inherited modifiers are concerned. The ocular system showed the strongest heritability (>60%) and is the only one for which a significant impact of variations at the major locus on the phenotype was found. Most skeletal or cardiovascular features displayed 40% to 60% heritability. The observation of a “Carter effect” in the cardiovascular phenotype provides further evidence of the existence of inherited modifiers and supports the high heritability of cardiovascular clinical features. The overall high heritability that was found is a robust result as it was replicated by two different methods (phenotypic correlation between relatives and Carter effect), and as heritability estimates might have been underestimated for two reasons. First, most clinical features were binary, which artificially reduces heritability, even more so for features with low prevalence [24]. Second, some features (mainly cardiovascular) are still evolving after 18 years of age, leading to heterogeneity between generations and thus decreasing parent–offspring correlations. Altogether, this study encourages further research to identify these modifiers. Finally, we developed an original approach to screen the influence of the major locus and provide a simple quantitative genetics model that can be adapted to study any rare multisystem genetic disease.

## Figures and Tables

**Figure 1 genes-11-00574-f001:**
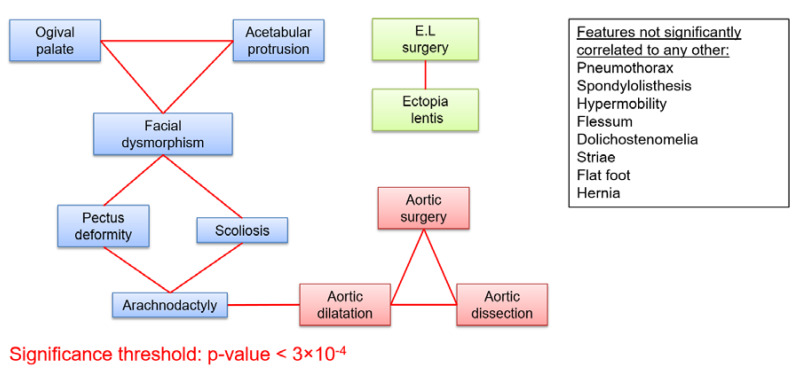
Cross correlations between 19 clinical features of MFS. Clinical features are represented as boxes. Red links between the boxes represent correlations significant after Bonferroni correction (*p* < 3 × 10^−4^). (E.L.: Ectopia lentis).

**Figure 2 genes-11-00574-f002:**
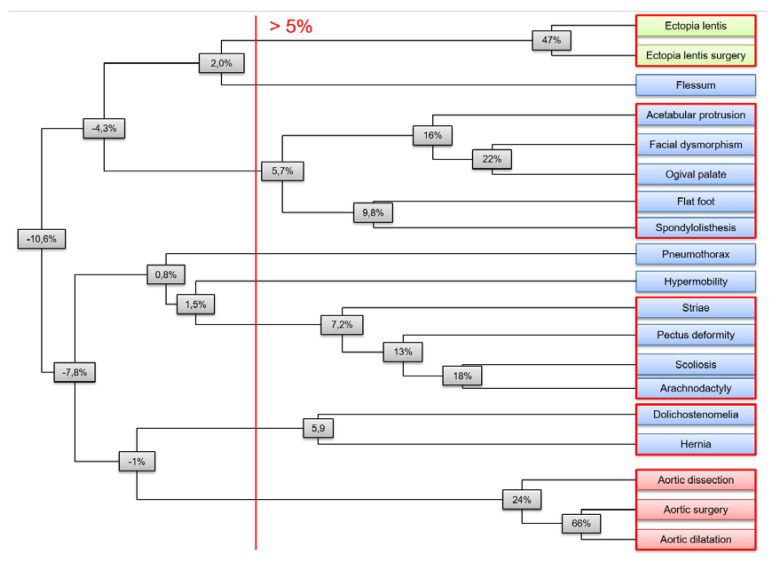
Agglomerative hierarchical clustering of clinical features of MFS. Agglomerative hierarchical clustering showed ocular (in green), cardiovascular (in red), and systemic clusters (in blue). Complete linkage clustering was chosen as linkage criteria, and the metric was Pearson correlations coefficient corrected for covariates. Node values indicate the minimal correlation between any couple of features within the corresponding subtree. On the right, the five clusters given by a 5% threshold are circled in red.

**Figure 3 genes-11-00574-f003:**
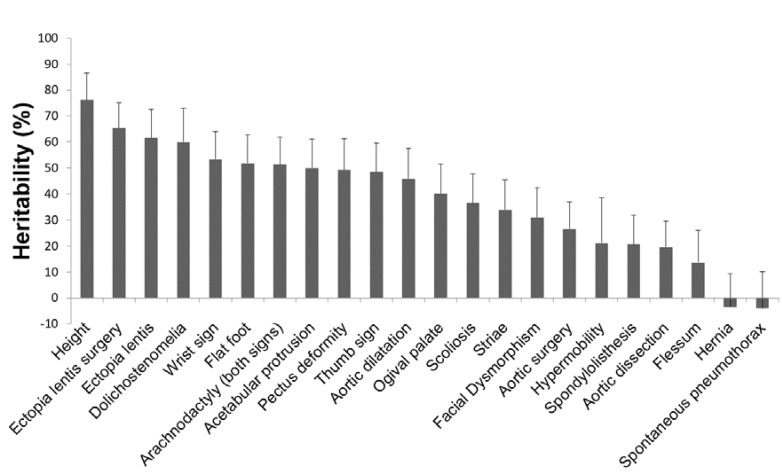
Heritability (H^2^) estimates of MFS clinical features. Heritability estimates were based on familial aggregation of clinical features among first-degree relatives. Error bars show standard deviations (SDs) and rely on asymptotic SDs of parent–offspring and siblings correlations provided by S.A.G.E. (Statistical Analysis for Genetic Epidemiology).

**Figure 4 genes-11-00574-f004:**
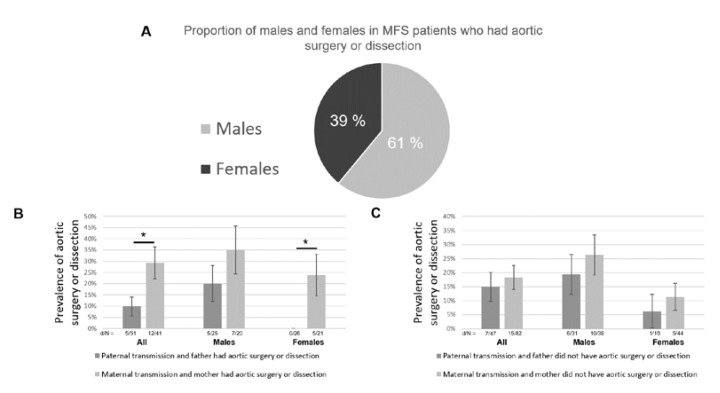
The Carter effect on the cardiovascular phenotype of MFS. (**A**) 61% of patients that had aortic surgery or dissection were males (*p* = 4.9 × 10^−5^). (**B**) Among MFS patients whose transmitting parent had aortic surgery or dissection, the prevalence of aortic surgery or dissection was significantly higher in the case of maternal transmission (*p* = 0.018). Standard deviations (SDs) are represented by error bars. d/N: number of patients with aortic surgery or dissection/sample size. (**C**) Among MFS patients whose transmitting parent did not have aortic surgery or dissection, the prevalence of aortic surgery or dissection was slightly higher in the case of maternal transmission (non-significant). SDs are represented by error bars. d/N: number of patients with aortic surgery or dissection/sample size. *: *p* < 0.05.

**Table 1 genes-11-00574-t001:** Phenotypic correlations between first degree relatives and heritability estimates. Phenotypic correlations were computed on S.A.G.E (Statistical Analysis for Genetic Epidemiology) after correction for age and sex.

	Parent–Offspring		Siblings		Heritability (H^2^%, ± S D)
	Correlation (%, ± SD)	Sample Size (n)	Correlation (%, ± SD)	Sample Size (n)	
Height	37.4 ± 6.7	193	38.8 ± 8.0	155	76.2 ± 10.4
Ectopia lentis surgery	35.8 ± 6.5	183	29.5 ± 7.4	162	65.3 ± 9.9
Ectopia lentis	32.5 ± 6.6	184	29.1 ± 8.9	163	61.6 ± 11.1
Dolichostenomelia	27.8 ± 9.2	142	32.3 ± 9.2	118	60.1 ± 13.0
Wrist sign	21.5 ± 6.9	193	31.8 ± 8.3	173	53.3 ± 10.8
Flat foot	22.0 ± 7.0	185	30.0 ± 8.5	163	51.9 ± 11.0
Arachnodactyly	19.7 ± 6.8	192	31.6 ± 8.2	170	51.3 ± 10.6
Protrusio acetabulae	24.6 ± 7.5	164	25.3 ± 8.3	132	50.0 ± 11.2
Pectus deformity	13.1 ± 8.0	187	36.2 ± 9.1	163	49.3 ± 12.1
Thumb sign	19.8 ± 6.9	188	28.6 ± 8.8	166	48.4 ± 11.2
Aortic dilatation	22.8 ± 8.2	184	23.0 ± 8.6	147	45.8 ± 11.8
Ogival palate	27.2 ± 6.5	187	13.0 ± 9.4	166	40.1 ± 11.4
Scoliosis	20.2 ± 7.1	185	16.4 ± 8.6	163	36.6 ± 11.2
Striae	12.9 ± 7.0	192	21.0 ± 9.2	171	33.9 ± 11.6
Facial dysmorphism	11.0 ± 6.8	188	20.0 ± 9.4	162	31.0 ± 11.6
Aortic surgery	6.8 ± 6.1	265	19.8 ± 8.2	222	26.6 ± 10.2
Hypermobility	−8.1 ± 11.3	183	29.3 ± 13.2	168	21.2 ± 17.4
Spondylolisthesis	15.3 ± 7.6	154	5.4 ± 8.3	124	20.7 ± 11.2
Aortic dissection	−2.5 ± 7.1	265	22.0 ± 7.2	222	19.5 ± 10.1
Flessum	11.1 ± 7.9	174	2.6 ± 9.6	166	13.6 ± 12.4
Hernia	−0.8 ± 6.9	177	−2.7 ± 10.9	157	−3.5 ± 12.9
Pneumothorax	−2.2 ± 6.6	192	−1.8 ± 12.6	172	−4.0 ± 14.2

**Table 2 genes-11-00574-t002:** Phenotypic correlations between distant relatives.

	Parent–Offspring (N ≈ 260)	Siblings (N ≈ 190)	Uncle–Nephew (N ≈ 170)	Cousins (N ≈ 95)
Arachnodactyly	23% ± 6.2	29% ± 7.5	8.5% ± 9.1	16% ± 19
Ectopia lentis surgery	41% ± 5.5	36% ± 7.4	19% ± 14	45% ± 12
TAA	25% ±6.4	26% ± 7.7	30% ± 9.1	13% ± 18

Estimate (%) ± SD. Phenotypic correlations between distant relatives provided an insight into the influence of the major locus. The high correlation between cousins for ectopia lentis surgery reflected the influence of the major locus FBN1 on ocular phenotype and is consistent with the literature.

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
