# Peer review of "Quantifying the Genetic Basis of Marfan Syndrome Clinical Variability"

_genes, 2020, doi:10.3390/genes11050574_

Round 1

Reviewer 1 Report

This article addresses quantification of inherited modifiers in the clinical variability of Marfan syndrome (MFS). This is a challenging approach, but the authors have the patient cohort and expertise available to start analyzing this aspect. Strong correlations between features within the same organ system were observed, suggesting common underlying determinants. However, clinical features in different systems were largely uncorrelated. A significant contribution of the major gene locus on the phenotypic variance for ectopia lentis was reported. Evidence was presented for the MFS cardiovascular phenotype, which supports a polygenic model for MFS cardiovascular variability. The study demonstrates that an important part of the phenotypic variability in MFS is under the control of inherited modifiers. Overall, this is an important and well-conducted study that will be very interesting to the clinical and scientific community. I only have some minor points:

  1. It is not clear how “ectopia lentis surgery” and “aortic surgery” can be heritable factors? Do the authors mean the need for those surgeries at a certain age? This aspect should be better explained.
  2. Line 52: Should read “MIM #154700” (because the number sign has a very specific meaning in MIM).
  3. Line 58: correct typo to “fibrillin-1”
  4. Line 302: correct typo to “symptoms”
  5. Line 368: ref 13 does not seem to be the correct reference for this statement.

Author Response

  1. It is not clear how “ectopia lentis surgery” and “aortic surgery” can be heritable factors? Do the authors mean the need for those surgeries at a certain age? This aspect should be better explained.

What is heritable is the fact that these events occured or not. This is now stated

  1. Line 52: Should read “MIM #154700” (because the number sign has a very specific meaning in MIM).

This has been corrected

  1. Line 58: correct typo to “fibrillin-1”

This has been corrected

  1. Line 302: correct typo to “symptoms”

Symptoms is written correctly

  1. Line 368: ref 13 does not seem to be the correct reference for this statement.

You are right the reference has been corrected

Reviewer 2 Report

This is a strong study addressing important clinical issues concerning genotype-phenotype correlations within Marfan syndrome. The idea is novel and the methods and analysis appear sound.

I would only have two suggestions:

1) Since the authors are looking at relationships between clinical symptoms, I am wondering why only simple correlation analyses were used and not something like multinomial regression where they could address more complex symptom interrelationships. I would either recommend additional analyses addressing more complex relationships or a reasonable justification for not doing so.

2) I would recommend further justifying use of the term "Carter effect" as opposed to simply "sex protective effects" as the two are not necessarily equivalent. Under the Carter effect, one would expect to see higher numbers of family members of the sex with higher threshold in families with greater genetic burden. Since genetic burden is not measured in these instances and it is the relationship between parent and child that are looked at, the authors may need to reconsider the use of the term, "Carter effect."

Author Response

I would only have two suggestions:

1) Since the authors are looking at relationships between clinical symptoms, I am wondering why only simple correlation analyses were used and not something like multinomial regression where they could address more complex symptom interrelationships. I would either recommend additional analyses addressing more complex relationships or a reasonable justification for not doing so.

We effectively did not perform a multinominal logistic regression because our objective was not to identify the relationship between one nominal dependent variable and one or more independent variables but to cluster the variables together ie to try to identify which variables may be under the control of the same set of genetic determinants. In the agglomerative hierarchical clustering we used Pearson correlations coefficient as metric, since it is classically used to identify charred genetic component between subjects or phenotypes even if charred environmental parameters may interfere.

2) I would recommend further justifying use of the term "Carter effect" as opposed to simply "sex protective effects" as the two are not necessarily equivalent. Under the Carter effect, one would expect to see higher numbers of family members of the sex with higher threshold in families with greater genetic burden. Since genetic burden is not measured in these instances and it is the relationship between parent and child that are looked at, the authors may need to reconsider the use of the term, "Carter effect."

To the best of our knowledge, the genetic burden relates to the presence in individuals of a number of rare to very rare variants. Using this terminology would alter our message that applies the molecular mechanism underlying the Carter effect: i.e.  presence of more numerous susceptibility polymorphisms in the mother. If reformulated using the current “variant” it would be worded as the “presence of more numerous frequent variants”

The comment probably reflects our lack of proper description of the “Carter effect” when it appears at first in the “Results section” (line 256 on page 9). Indeed, we provide a partial explanation only in the discussion section of the manuscript (line 375 on page 12). Therefore and for clarification, we modified the first two sentences as follows:

Finally, we looked at the influence of the sex of the transmitting parent on the cardiovascular phenotype and found “Carter Effect” [20]. The effect refers to the existence of a polygenic threshold model with sex dimorphism that postulates the presence of a greater number of susceptibility polymorphisms in affected carriers of one sex as compared to the other”.

Reviewer 3 Report

To the Authors:

  1. Can the authors assess cancers in the clinical variability (https://www.ncbi.nlm.nih.gov/pmc/articles/PMC5652471/).
  2. The authors should comment that the decision to remove a dislocated lens (ectopia lentis surgery) is more a function of the refractive skills of the surgeon. In most cases, good visual acuity can be achieved in patients with ectopia lentis if the pupils are dilated and a good refraction is performed. Thus, this reviewer would suggest only analyzing the variable ectopia lentis. Some surgeons will remove a dislocated lens without careful refraction. The authors should note that all patients with ectopia lentis surgery had ectopia lentis but not all patients with ectopia lentis undergo surgery.
  3. The authors should also comment on aortic dissection. My understanding is that it is a surgical emergency.
  4. The authors should add a short section in the Discussion and describe/cite a different autosomal dominant disorder that has been studied with respect to inherited genetic modifiers, especially if the study established the importance of modifiers.

Minor editorial comments:

  1. On line 38, the phrase ‘same system’ should be defined.
  2. On lines 79, the sentence beginning ‘The purpose…’ should start a new paragraph.
  3. On line 91, the phrase ‘quality…collected’ should be ‘of’.
  4. On line 99, the phrase ‘plain flat foot’ should be clarified.
  5. On line 111, this reviewer suggests replacing ‘, which’ with ‘that’. The phrase ‘that adapts’ should be ‘adapting the’. On line 112, ‘provides’ should be replaced with ‘providing’.
  6. On line 226, the word ‘had’ should be ‘has’ and the phrase ‘until now’ removed.
  7. On line 310, do the authors mean signs or symptoms?
  8. On line 321, the word ‘aortic’ should not be capitalized. The aortic features are ‘signs’ or ‘features.’
  9. On line 339, should the phrase be ‘and or’.
  10. On line 348, the word ‘comforted’ should be replaced.
  11. On line 350, the verb should be ‘decreases’.
  12. On line 362, the phrase should be ‘not sufficiently precise to permit conclusions’.
  13. On line 386, the word ‘between’ should be ‘among’.
  14. As the Conclusions may be shared without the benefit of the text, the authors should redefine ‘same systems’.

Author Response

To the Authors:

  1. Can the authors assess cancers in the clinical variability (https://www.ncbi.nlm.nih.gov/pmc/articles/PMC5652471/).

Cancer occurrence is not available in our database, and therefore are not able to assess clinical variability in this population. Sorry.

  1. The authors should comment that the decision to remove a dislocated lens (ectopia lentis surgery) is more a function of the refractive skills of the surgeon. In most cases, good visual acuity can be achieved in patients with ectopia lentis if the pupils are dilated and a good refraction is performed. Thus, this reviewer would suggest only analyzing the variable ectopia lentis. Some surgeons will remove a dislocated lens without careful refraction. The authors should note that all patients with ectopia lentis surgery had ectopia lentis but not all patients with ectopia lentis undergo surgery.

We agree with the reviewer. However, in our study, the 2 are closely related (figures 1; figure 2), so that the results obtained with ectopia lentis are similar to that obtained with surgery for ectopia lentis. This concordance is reassuring, because, as stated by the reviewer, the decision to do surgery is the decision of the surgeon, and the ability to detect ectopia lentis is dependent on the ophthalmologist.

  1. The authors should also comment on aortic dissection. My understanding is that it is a surgical emergency.

The sentence : “The life threatening complication is aortic dissection, a surgical emergency, which occurs in dilated aorta.” Has been added in the introducation

  1. The authors should add a short section in the Discussion and describe/cite a different autosomal dominant disorder that has been studied with respect to inherited genetic modifiers, especially if the study established the importance of modifiers.

We included 2 specific paragraphs in the discussion:

P11: It is interesting to note that despite the high heritability of these phenotypes, the identification of rare or common genetic modifiers remains a very difficult challenge. Our group, in a multiomic approach in a large cohort of 1070 clinically well-characterized FBN1marfan patients, identified nine modifier loci in polygenic model [26]. None of them has allowed the identification of the responsible variant(s) with certainty and the use of these results for genetic counselling is still far away. Few studies have measured the heritability of aortic aneurysm susceptibility. In particular, there are two Scandinavian studies using the measure of concordance between mono- and dizygotic twins from Swedish and Danish general population registries [27,28]. The phenotype studied was abdominal aortic aneurysm. The two studies give similar heritability results, 70% and 77% respectively and an influence of unshared environmental factors of 20-30%. These heritability values are significantly higher than our measurements for thoracic aortic aneurysm in Marfan patients, whereas environmental factors, such as smoking, are considered to be a determinant in the occurrence of abdominal aortic aneurysm. This result is all the more noteworthy since twin studies control for the influence of the shared environment, which intrafamilial correlation studies cannot do and therefore tend to overestimate heritability.

P12: There are few examples of studies measuring the heritability of different phenotypes associated with an autosomal dominant Mendelian disease. Sebbagh et al. were nevertheless able to measure the heritability of different phenotypes associated with neurofibromatosis type 1 in a study of phenotypic correlations in 275 multiplex French NF1 families. As in our study, patterns of familial correlations indicated a strong heritability with no apparent influence of the constitutional NF1 mutation [30].

Minor editorial comments:

  1. On line 38, the phrase ‘same system’ should be defined.

This has been done: We found strong correlations between features within the same system (i.e. ophthalmology vs. skeletal vs. cardiovascular)

  1. On lines 79, the sentence beginning ‘The purpose…’ should start a new paragraph.

Done. The sentence is now: This ensures a high level of homogeneity and quality of data collection”.

  1. On line 91, the phrase ‘quality…collected’ should be ‘of’.

Done. The sentence is now: This ensures a high level of homogeneity and quality of data collection”.

  1. On line 99, the phrase ‘plain flat foot’ should be clarified.

Changed for “flat feet”

  1. On line 111, this reviewer suggests replacing ‘, which’ with ‘that’. The phrase ‘that adapts’ should be ‘adapting the’. On line 112, ‘provides’ should be replaced with ‘providing’.

done

  1. On line 226, the word ‘had’ should be ‘has’ and the phrase ‘until now’ removed.

done

  1. On line 310, do the authors mean signs or symptoms?

Signs, this has been changed

  1. On line 321, the word ‘aortic’ should not be capitalized. The aortic features are ‘signs’ or ‘features.’

changed

  1. On line 339, should the phrase be ‘and or’.

changed

  1. On line 348, the word ‘comforted’ should be replaced.

It was changed for strengthened

  1. On line 350, the verb should be ‘decreases’.

corrected

  1. On line 362, the phrase should be ‘not sufficiently precise to permit conclusions’.

corrected

  1. On line 386, the word ‘between’ should be ‘among’.

corrected

  1. As the Conclusions may be shared without the benefit of the text, the authors should redefine ‘same systems’.

Corrected :(i.e. ophthalmology vs. skeletal vs. cardiovascular)